# Exploring the perceptions and experiences of community rehabilitation for Long COVID from the perspectives of Scottish general practitioners' and people living with Long COVID: a qualitative study

Kay Cooper ,[1] Edward Duncan ,[2] Erin Hart-Winks,[1] Julie Cowie ,[3] Joanna Shim ,[1] Emma Stage ,[1] Tricia Tooman,[2] Lyndsay Alexander ,[1] Alison Love,[4] Jacqui H Morris ,[5] Jane Ormerod,[4] Jenny Preston ,[2,6] Paul Swinton [1]

KC and ED are joint first authors.

For numbered affiliations see end of article.

**Correspondence to**
Professor Kay Cooper;
k.cooper@rgu.ac.uk

## ABSTRACT

**Objectives** To explore the experience of accessing Long COVID community rehabilitation from the perspectives of people with Long COVID and general practitioners (GPs).

**Design** Qualitative descriptive study employing one-to-one semistructured virtual interviews analysed using the framework method.

**Setting** Four National Health Service Scotland territorial health boards.

**Participants** 11 people with Long COVID (1 male, 10 female; aged 40–65 (mean 53) and 13 GPs (5 male, 8 female).

**Results** Four key themes were identified: (1) The lived experience of Long COVID, describing the negative impact of Long COVID on participants' health and quality of life; (2) The challenges of an emergent and complex chronic condition, including uncertainties related to diagnosis and management; (3) Systemic challenges for Long COVID service delivery, including lack of clear pathways for access and referral, siloed services, limited resource and a perceived lack of holistic care, and (4) Perceptions and experiences of Long COVID and its management, including rehabilitation. In this theme, a lack of knowledge by GPs and people with Long COVID on the potential role of community rehabilitation for Long COVID was identified. Having prior knowledge of rehabilitation or being a healthcare professional appeared to facilitate access to community rehabilitation. Finally, people with Long COVID who had received rehabilitation had generally found it beneficial.

**Conclusions** There are several patient, GP and service-level barriers to accessing community rehabilitation for Long COVID. There is a need for greater understanding by the public, GPs and other potential referrers of the role of community rehabilitation professionals in the management of Long COVID. There is also a need for community rehabilitation services to be well promoted and accessible to the people with Long COVID for whom they may be appropriate. The findings of this study can be used by

## STRENGTHS AND LIMITATIONS OF THIS STUDY

⇒ The issue of accessing Long COVID community rehabilitation was explored from the perspectives of both potential referrers and recipients of community rehabilitation. One researcher conducted all interviews, ensuring consistency in their conduct.

⇒ Data were analysed and interpreted by multiple researchers, including people with Long COVID.

⇒ The study is limited by the small and predominantly female sample, largely drawn from health boards adopting a similar approach to Long COVID rehabilitation.

those (re)designing community rehabilitation services for people with Long COVID.

## INTRODUCTION

Long COVID (LC) in adults is a multisystemic condition described as signs and symptoms that develop during or after an infection consistent with COVID-19 and continue for more than 12 weeks, which cannot be explained by an alternative diagnosis.[1] While global prevalence remains unclear, the WHO estimated that 34 million people may have experienced LC by 2022.[2] In the UK, 1.9 million people (2.9% of the population) had self-reported LC as of March 2023.[3]

Rehabilitation for people with LC is recommended in practice guidelines[1 4] and research,[5–7] with UK guidelines recommending personalised and multidisciplinary services.[1] Rehabilitation can be defined as intervention/s aimed at optimising function and reducing disability.[8] By definition,

rehabilitation is multidisciplinary and applied to a range of health conditions. While it is increasingly recognised that the signs and symptoms of LC share commonalities with other chronic conditions, including fibromyalgia and myalgic encephalomyelitis (ME)/chronic fatigue syndrome (CFS)[9] and that condition-specific rehabilitation services may further fragment already scarce resources,[10] different approaches to LC rehabilitation have emerged. In England, service provision for LC has involved the development of specialised clinics, usually comprising multidisciplinary teams of medical, allied health and psychology professionals.[11] In Scotland, LC rehabilitation is delivered in community (non-hospital) settings by generalist allied health professionals, with local variation in models of service delivery.[12]

People with LC have reported barriers to accessing healthcare and difficulties in navigating disjointed healthcare services.[12–15] General practitioners (GPs) (synonymous with family physicians/doctors) are the first point of contact for people with LC and have an important role in ensuring that people receive appropriate treatment[16] and referral to specialist services, including community rehabilitation, when needed.

There is a growing body of research on the lived experience of people with LC[15 17 18] including barriers to accessing healthcare[13 14] Until recently, however,[15] none had focused on community rehabilitation. There is also a growing body of research on the role of the GP in providing care to people with LC, including the challenges of diagnosis, medical management and difficulty accessing community services such as rehabilitation.[19 20]

The current study aimed to add to the growing body of knowledge by exploring people with LC and GPs' perspectives of accessing LC community rehabilitation in the Scottish context, due to the approach to LC in this setting being distinct from other parts of the UK.[8 12] In this study, community rehabilitation was defined as rehabilitation[10] delivered by any appropriate healthcare professional in a community setting; typically in clinics (outptaient/ambulatory) and people's homes.

This work was embedded within a larger realist evaluation of LC community rehabilitation in Scotland, which identified low numbers of people with LC receiving community rehabilitation services.[15]

This study aimed to address two questions: (1) what are the perceptions and experiences of people with LC on accessing rehabilitation for LC? and (2) what are GPs' perceptions and experiences of managing people with LC presenting with symptoms of LC that may be suitable for rehabilitation?

## METHODS
### Study design
This was a qualitative descriptive study employing semi-structured interviews with a convenience sample of people with LC and GPs in 4 of the 14 Scottish territorial health boards. The study followed an a priori protocol (online supplemental file 1) and is reported in keeping with the Consolidated criteria for Reporting Qualitative research (online supplemental file 2).[21]

### Participants
The four Scottish health boards were chosen for the realist evaluation study based on variation in population and accessibility (Scottish Government Urban Rural Classification 2020),[22] LC prevalence and rehabilitation service delivery models. Two health boards were offering an integrated LC rehabilitation service (ie, integrated within existing community rehabilitation pathways), one had recently launched a dedicated LC community rehabilitation service, and one pre-existing dedicated LC service was closed to new referrals due to an increased referral rate combined with reduction in funding and therefore inability to staff the service (personal communication).

A convenience sample of people with LC residing in the four health boards was recruited via social media accounts (Facebook; X (formerly Twitter)) of the research team and their institutions and by Long COVID Scotland, a volunteer-led charity run by people with LC. Inclusion criteria for people with LC were community-dwelling (ie, not currently hospitalised); aged 18 or over; experiencing symptoms of LC (with or without a positive COVID-19 test), and experience of accessing or attempting to access healthcare services for possible rehabilitation. Those interested in the study contacted the research team, were sent detailed study information and provided informed consent (audio recorded) prior to taking part.

A convenience sample of GPs was recruited by email invitation circulated to eligible GP practices by the National Health Service Research Scotland Primary Care Network coordinator. Inclusion criteria were GP in a practice within one of the four eligible health boards and experience of patients with probable LC who may be suitable for rehabilitation. Several recruitment reminders were issued. We aimed to recruit 12–20 people with LC and 8–20 GPs.

### Data collection
Interview topic guides (online supplemental file 3) were developed. While not formally pilot tested, they were refined by the research team in consultation with people with lived experience of LC, and informed by initial findings from our realist evaluation study[15] and the wider literature in the field.

During the period June 2022–January 2023, semistructured online interviews (Microsoft Teams) and one telephone interview were conducted by one research assistant (EH-W), who received training and supervision from KC and ED. Online interviews have been used for some time.[23] While they became a necessity during the COVID-19 pandemic[24] their potential is increasingly recognised.[25] In this study, they facilitated participation across a large geographical area and flexibility for participants while avoiding unnecessary social contact for people with LC. No repeat interviews were undertaken. GPs were either

in their workplace or homes when interviewed; people with LC were all at home. The female research assistant was a qualified nurse and had worked in critical care from the start of the COVID-19 pandemic (March 2020) until December 2021. She had no prior relationship with the GP practices or participants who all understood that she was employed as a research assistant.

Interviews lasted 17–47 min (mean 23±7.8 min) and were audio recorded. No other individuals were present during the interviews. No field notes were taken. Interviews with people with LC were transcribed by an external transcription service. Due to the homogeneity of the GP interviews, these were not transcribed but instead were listened to on multiple occasions by two researchers, a method informed by the wider literature.[26 27] Neither transcripts nor audio files were returned to participants for comment or correction.

## Data analysis
Data were analysed using the framework method (Gale et al),[28] which proposes a matrix-based format to facilitate the sharing and management of data as a team, and is widely used in applied health research. Familiarisation with the data began by reading and re-reading the transcripts (people with LC) and listening/relistening to the recordings (GPs), making analytical notes that informed the 'working analytical framework' for each participant group.[28] Although line-by-line coding is common in qualitative research, it is also possible to develop a framework without engaging in explicit coding[29]; due to the small scale and nature of the data we adopted the latter approach.

The framework method was used to construct matrices in Microsoft Excel, enabling the data to initially be summarised into broad categories. This was led by EH-W, in close consultation with two experienced qualitative researchers (KC and ED). The charted data were then analysed by interpreting within and between participants to identify concepts, which were subsequently grouped into themes, and finally, overarching themes consisting of data from both participant groups. This was an iterative process involving multiple researchers (KC, ED, JC, TT, JS and ES) and the wider study team, with subsequent refinement until there was consensus that the data had been comprehensively analysed.

As with our previous study,[15] we did not seek participant feedback directly on the findings. However, we presented the findings in a webinar attended by health professionals and people with LC, who endorsed the findings, reporting that the analysis reflected their personal experiences.

## Patient and public involvement
Two members of the public with lived experience of LC were core members of the study team (AL and JO). Both contributed throughout the study and were integral to its success, helping with design and identification of important issues to explore. They codeveloped study materials and interview topic guides and contributed to analysis and interpretation of findings.

## RESULTS
### Participant characteristics
25 people with LC expressed an interest in taking part in the study. Of these 25, 1 did not meet the inclusion criteria for this study but was invited to take part in the realist evaluation. 13 people with LC decided against taking part (reasons unknown). 16 GPs expressed an interest, of whom 3 did not proceed to interview due to logistical challenges. Therefore, 11 people with LC (1 male, 10 female; aged 40–65 (mean 53)) and 13 GPs (5 male, 8 female) provided informed consent and took part in an interview. Figure 1 displays the numbers of participants recruited from each of the four health boards and table 1 describes participants' demographics. No participants who consented withdrew from the study.

### Findings
Framework analysis resulted in four overarching themes: (1) The lived experience of LC; (2) The challenges of an emergent and complex chronic condition; (3) Systemic challenges for LC service delivery and (4) Perceptions and experiences of LC and its management, including rehabilitation. Table 2 identifies the data and participant groups that contributed to each of these themes. Throughout the manuscript, participants are referred to by a unique, anonymous identifier (health board A–D, followed by PwLC (person with LC) or GP, followed by participant number).

#### The lived experience of LC
People with LC described the negative impact of a range of physical and psychological symptoms on their quality of life and physical and social participation, including employment. Many described the impact as life changing and a source of great frustration. Fatigue was common, as was poor mental health, with symptoms of anxiety and depression described by some participants, with both pharmacological and psychological support being sought.

> I was desperate, I just had no quality of life at all. I couldn't speak to my friends for coughing, couldn't look after my family because I had no energy, couldn't get out the house, I was housebound for months. It was just rubbish. [APwLC05]

Employment was a common source of concern. Although some participants had attempted a phased return to work, there was generally a perceived lack of support for returning to work after a lengthy LC-related absence, with a need for vocational rehabilitation and practical support to alleviate financial anxiety identified. Presenteeism was also identified as an issue for some people with LC who were at work but not able to fully function due to their symptoms.

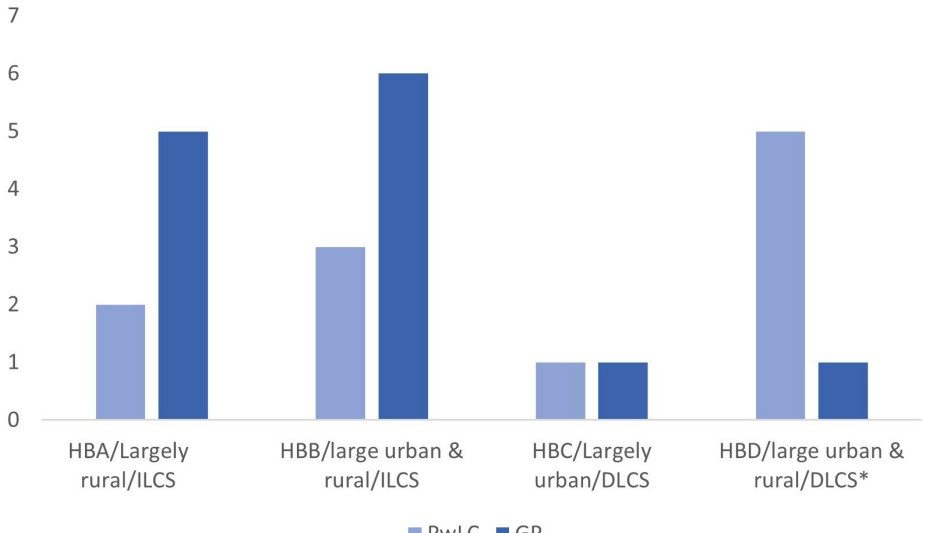

**Figure 1** Number of general practitioners (GP) and people with Long COVID (PwLC) recruited from each health board. *closed to new referrals at the time of data collection. HBA, Health Board A; HBB, Health Board B; HBC, Health Board C; HBD, Health Board D; ILCS, Integrated Long COVID Service; DLCS, Dedicated Long COVID Service

Fortunately, I've had it, been off, but then went back to work. I've not stayed off. Maybe that's a problem as well, I've not stayed off long term and listened to my body. I've went back and really threw myself back into my workplace and then suffered in my days off. [CPwLC01]

### The challenges of an emergent and complex chronic condition

Many participants had presented to primary care seeking confirmatory tests and onward referral due to their complex symptoms and experienced frustration when GPs were reluctant to diagnose LC. GPs, however, spoke

of the challenges associated with providing a diagnosis in the absence of a diagnostic test, and the limited treatment options available to them.

I don't know if we will move away from handing out the diagnosis of Long COVID. Because to be honest, I don't often suggest it to the patient as a diagnosis because our options for management are so minimal. So, I tend to, if the patient thinks they've got it, work through that with them. [AGP01]

GPs drew similarities between LC and conditions such as CFS/ME and fibromyalgia, particularly the uncertainties associated with diagnosis and management, conveying the need for a long-term conditions service inclusive of LC.

I think we are talking about Long COVID now because everyone is looking at COVID, which is great. But I think it's not the only kind of post-[illness] treatment type problem that we have nowhere to send people. And I think it's a problem with the health system. [BGP01]

People with LC also recognised the uncertainties associated with LC. However, many felt they were not heard or believed by healthcare professionals and the wider public and expressed a desire for their symptoms to be validated. Participants reported living with an invisible illness, often attributed to a lack of understanding about how LC affects a person, and a feeling that they were being treated like a malingerer.

There's definitely been a lot more media which is very helpful…there was a middle part where people were quite scathing because in the first part I remember friends saying 'I don't know anybody else that's

| Table 1 | | Participant demographics | |
|---|---|---|---|
| **GPs** | **n (%)** | **People with Long COVID** | **n (%)** |
| Sex | | Sex | |
| Male | 5 (38) | Male | 1 (9) |
| Female | 8 (62) | Female | 10 (91) |
| Years experience general practice | | Employment status | |
| | | Employed | 4 (36) |
| 1–10 | 4 (31) | Long-term sick leave | 4 (36) |
| 11–20 | 4 (31) | Left employment due to Long COVID | 3 (27) |
| >20 | 3 (23) | | |
| NR | 2 (15) | | |
| | | Comorbidities | |
| | | Yes* | 5 (45) |
| | | No | 6 (55) |

*Comorbidities included respiratory and thyroid conditions, and myalgic encephalomyelitis.
GP, general practitioner; NR, not reported.

**Table 2** Overview of thematic analysis

| Overarching themes | Themes | Concepts |
|---|---|---|
| A: The lived experience of LC | **People with LC**<br>Impact of LC on daily life | **People with LC**<br>Psychological impact of LC<br>Physical symptoms of LC<br>Impact of LC on quality of life<br>Impact of LC on ability to work |
| B: The challenges of an emergent and complex chronic condition | **GP**<br>LC as a chronic condition<br>**People with LC**<br>LC as a new and unknown condition | **GP**<br>Challenges and uncertainty associated with diagnosis<br>Complexities of LC and similarities with chronic conditions (ME/FM/CFS/postviral)<br>System challenges associated with chronic conditions<br>**People with LC**<br>Stigma associated with LC<br>Invisibility of LC<br>Lack of validation<br>Complex symptoms and lack of diagnostic/confirmatory tests<br>Lack of ownership by any given discipline and related difficulty getting referred to secondary care<br>Need for self-advocacy among people with LC |
| C: Systemic challenges for LC service delivery | **GP**<br>LC management in primary care<br>**People with LC**<br>Barriers and facilitators to accessing healthcare support for LC | **GP**<br>Current pathways/lack of pathways<br>Resource issues<br>Training needs<br>Safety netting: need for tests and investigations<br>**People with LC**<br>Lack of referral pathways and access to GP and/or secondary care<br>Lack of available resources with long waiting lists and limited support<br>Need for joined up and person-centred care, especially follow-up on results of tests and investigations<br>Apparent inequities across health boards |
| D: Perceptions and experiences of LC and its management, including rehabilitation | **GP**<br>Perceptions of LC<br>**People with LC**<br>Healthcare professionals' knowledge and attitudes<br>**GP**<br>LC Rehabilitation<br>**People with LC**<br>Experience of LC services | **GP**<br>Knowledge and beliefs about LC prevalence<br>Perceptions of symptom presentation<br>Knowledge and beliefs about risk factors<br>Knowledge and beliefs about subgroups affected by LC<br>Work-related issues and need for vocational rehabilitation<br>Psychological impact and need for peer support<br>Beliefs and values relating to LC<br>**People with LC**<br>Experiences of support available<br>Importance of validation from primary care<br>Perceptions of HCPs lack of knowledge and understanding of LC<br>Role of self-management in LC<br>Role of pharmacology in LC<br>**GP**<br>Role of medical staff in LC Rehabilitation<br>Role of allied health professionals in LC Rehabilitation<br>Rehabilitation needs<br>**People with LC**<br>Experience of support from allied health professionals: education and symptom management<br>Perceptions of usefulness of online resources (mixed views of social media)<br>Views on private healthcare<br>Perceptions and experience of third sector support<br>Perceptions of peer support<br>Perceptions and experience of occupational health support |

CFS, chronic fatigue syndrome; FM, fibromyalgia; GP, general practitioners; LC, Long COVID; ME, myalgic encephalomyelitis/encephalopathy.

got Long COVID', and that in itself felt a judgement, but then we hit a middle part where loads of folk were getting COVID but they'd been vaccinated or their bodies just dealt with it differently and they weren't ill, so then there was a huge period of judgement came out, 'well, so-and-so had it and they're fine'. [BPwLC06]

Many people with LC felt their symptoms were not taken seriously enough to gain access to secondary care. Some participants' approach to this was to take control of their own health, learning to advocate for themselves and researching symptoms and self-management interventions online.

Oh, I've turned to Twitter…there is kind of a few people on Twitter that I follow that have been really good and kind of published research papers…I go back to the GP and ask about stuff, so I'm just having to kind of search for it myself…So you're not only having to deal with the illness you have to kind of then navigate kind of like, where am I going to get help from? [DPwLC02]

### Systemic challenges for LC service delivery

Healthcare system-related challenges included the lack of clear pathways for accessing services, siloed specialised services, limited resources (resulting in long waiting times) and the lack of holistic care, with limited resources identified as one of the main barriers. People with LC described these challenges as contributing to communication breakdowns, a lack of accountability over chronic conditions and the need for self-advocacy among people with LC.

I feel that if there'd been a more joined up approach, somebody would have been like "wait a minute, she can't be on that for four months on that dose" … because my doctor at the time had a consult thing online. So all I would do is type in what my symptoms still were and the chemist would deliver drugs. I didn't physically talk to anybody. So, that was obviously a huge failing, and I think doctors really realised that should never have happened but they had such shortages and still have such shortages [BPwLC05]

GPs reported a lack of pathways and processes for managing patients with LC, including identifying and tracking patients along the care trajectory and establishing consistent management approaches. The nature of the emergent condition made it challenging to identify appropriate services to refer patients on to. GPs reported that patients seldom met the criteria for existing rehabilitation programmes (eg, cardiac/respiratory). They also reported that they often carried out tests and investigations to exclude other differential diagnoses as a means of 'safety-netting', but some were mindful that this could delay rehabilitation and recovery.

There is a risk that we hold up rehabilitative inputs until we fully investigated things and we are entirely assured that there's nothing going on. So, we should perhaps be blending things a little better. [BGP02]

People with LC experienced difficulty accessing both primary and secondary care. GPs were aware of the significant pressure secondary care were facing which often resulted in long waiting times for specialties. As a result, patients often re-presented to primary care for ongoing issues leaving GPs with limited options.

I didn't have the energy to argue with the receptionist at the GP surgery, and that's the honest truth, I just didn't have it in me to phone and try and explain it all again. So, I lost nineteen pounds in four weeks because I just couldn't eat because I felt sick. But even that wasn't worth the battle I would have to get past a GP receptionist. [APwLC05]

We were getting a bit frustrated referring patients to secondary care for help. There wasn't much coming through. They were already dealing with backlog enough and they're getting piled up with these other things happening. I understand their limitations, entirely…I don't think that any I have referred [to respiratory or cardiovascular] have been seen yet. … [CGP03]

People with LC reported an awareness of variation in access to LC rehabilitation, and the apparent inequity compared with LC clinics in England. Some participants sought private healthcare and reported subsequent improvements in their condition. People with LC emphasised the need for a more joined-up service to improve communication and coordination of care. They wanted healthcare professionals to focus on the condition as a whole and not just specific symptoms.

I just feel sometimes having that one person, like I know a lot of people have a consultant that they go to and that's the person that they speak to, or the centre that they go to, for support. There's a group of people perhaps that they deal with, but they get it. They're a familiar face and that makes sense. But you feel a bit of a pariah, to be honest, with Long COVID. [BPwLC06]

GPs also acknowledged the importance of a more holistic approach to LC care including psychological and physical support. They suggested the need for integrated multidisciplinary management providing support for a complex range of symptoms.

If there was a secondary care service set up to see a certain number of patients per day then presumably they would probably allow further, probably allow longer appointments and would be more of a multidisciplinary approach so that patients would have a bit more time to kind of unpick everything that's going on because there definitely is, probably a kind of

you know, whole sort of biopsychosocial thing going on here [AGP02]

## Perceptions and experiences of LC and its management, including rehabilitation

### GPs knowledge, attitudes and perceptions of LC

In health boards where LC management was integrated into existing community services with less clear pathways to rehabilitation (health boards A and B), there was a perceived lack of demand by GPs for LC-specific services. GPs reported low numbers (1–6 per week) presenting for support with symptoms of LC.

Maybe 10–12 [Long COVID patients have presented] in total. But I don't know whether they're all coming to us. They might be just suffering in silence. [CGP03]

Patients were commonly reported to be of working age and were perceived to be fit and healthy prior to COVID-19 infection. Some GPs also reported that females were more commonly presenting with symptoms associated with LC than males.

Some GPs attributed the low patient numbers to natural resolution and the ability to self-manage on the basis that patients were not reconsulting or requesting further sick lines.

Most people are pretty sensible and they know, they are educated and I am finding that patients that are coming with long covid symptoms, or being quite certain that they have long covid they have educated themselves about it so, quite often actually they would come and they would sometimes know much more than the doctor about it because they have done their own research about it and they will likely know that there is nothing else [BGP05]

However, some GPs and most people with LC believed that people were not presenting to primary care because they believed there was no support available.

A lot of patients don't necessarily consult because probably they are seeing things in the media and things, you know aware that there aren't particular treatments. So, they just think it's par for the course that they feel like that. [AGP02]

I stopped contacting the GP because I just feel I'm wasting their time [APwLC03]

People with LC recognised that GPs were under pressure and some attributed their reduced attendance at GP practices to feeling like they were an additional burden on the healthcare system. They spoke of withholding information related to their symptoms due to an awareness that GPs were time pressured and expressed concerns associated with ensuring appointments were productive.

I know there are things that I haven't raised with a GP because I'm aware that they're time pressured, I've raised about ten symptoms already in my consultation

with them and I know I've got another three sitting on my list, but I can't bring that into the situation… and I'm potentially sitting on stuff that I should have discussed [BPwLC06]

People with LC also reported concerns over GPs apparent lack of specialist LC knowledge.

But the last time I went to the GP they said, 'we find people with Long COVID know more about it than we do.' And I thought that doesn't really fill you with great enthusiasm. [DPwLC04]

Most GPs acknowledged that LC has impacted people living with the condition and conveyed a need for education associated with recovery and returning to work. They reported that patients felt pressured to return to work without appropriate support in place to assist workplace integration. The need for psychological support was also expressed by both GPs and People with LC. Peer support was identified as a useful resource for people with LC, where they can be supported by others experiencing similar symptoms.

It's just reassuring to know that you're not alone in this. Misery loves company, and it's good to know that there are other people who have this, because otherwise it would become kind of depressing. And it helps put things in perspective, that you know that as bad as you feel someone else is probably feeling worse. [DPwLC06]

A minority of GPs did not perceive the need for a specialised LC service. Some GPs referred to a scepticism among their colleagues about LC in general:

It's not only for people, it's also for the medical professionals to believe as well that this is a problem. I think there is still some scepticism among medical professionals as well, still, about this being accepted and treated. [CGP03]

One GP described themselves as being slightly cynical about LC and wary of the need for rehabilitation, suggesting that patients should 'wait it out' and they will get better, likening LC and its impact to 'influenza.

I think it's exactly like flu. The same applies in flu. You get lots of people that get it. Most people are not terribly well with it, few people get flu without knowing they've had it. Some people recover quickly, some take a longer time to recover and some die [AGP04]

### Perceptions and experiences of rehabilitation and other LC services

A range of perceptions regarding LC rehabilitation was held by both participant groups. Some people with LC lacked knowledge of the potential role of rehabilitation professionals in supporting people with LC. This view

was shared by some GPs who reported a limited understanding of the role of rehabilitation in the management of LC.

> Physio, I can't really see much of a role. But that could be my lack of knowledge about it because with the patients I have spoken to it's not really so much of a physical thing, it's not like a particular joint pain as such that they would benefit from a physio. It's more the kind of cognitive aspect, maybe an OT, but I don't know what they would add. [BGP05]

Some people with LC had enough knowledge of rehabilitation to request referral or refer themselves to rehabilitation services where this was an option. In most cases these participants were on waiting lists and had been for some time. Although we did not record participants' job titles (for those in employment), several participants in this category disclosed during the interviews that they were healthcare professionals, with prior knowledge of, or colleagues working in, rehabilitation services. A third group of people with LC reported a need for LC rehabilitation services in their health board area, suggesting a lack of services and/or their promotion.

The three participants who had received rehabilitation for LC were generally positive about their experience, reporting benefits from specific professions (eg, physiotherapy, speech and language therapy) and interventions (eg, breathing exercises), with information and advice on LC and symptom management, particularly the use of pacing, being highly valued.

> The biggest help was speaking to Speech and Language… she gave me lots of information that was very interesting, and lots about the biology of what's going on with my [laryngeal] spasms. [DPwLC04]

> Just having somebody to help you manage what that should look like, what is too much, because you can read about pacing, you can chat about it online with other people with Long COVID, but trying to get a model that fits for you as an individual is actually really hard without support. [DPwLC06]

This contrasted with the views of people with LC of generic self-management booklets, which were commonly reported as lacking person-centredness.

> When I did refer myself to the [specialist] team, I got a booklet, a massive booklet through the post, that says this that and the other. But it's such an individual, highly differentiated set of symptoms that any one person can have, just none of it was particularly relevant to me. [DPwLC01]

One participant had sought informal rehabilitation from a personal trainer but felt that it was not helpful due to its intensity, feeling that the trainer did not have the requisite LC knowledge to support individuals effectively.

> Just with having this discussion it's like a wee light bulb moment that I'm having, that I'm thinking I've

tried the PT [personal training], it was too intense, threw the towel in. [DPwLC01]

Despite the lack of knowledge of the potential role of rehabilitation demonstrated by some GPs, most reported that they would engage with a dedicated LC rehabilitation service, as it would provide an onward referral route for patients, particularly as they commonly reported being limited by secondary care referral criteria. Several GPs felt that a multidisciplinary team approach could be beneficial for their patients and could provide the validation that patients needed. Some also believed that earlier pulmonary rehabilitation could contribute to better functional recovery.

> I think they feel quite isolated actually and I think it would be useful even if objectively…there's not a huge improvement. I think psychologically it would be really important for them. Someone to believe them, to see what's happening, and just thinking someone's looking out for them. [BGP01]

People with LC reported accessing a range of other LC services and sources of support. For some, occupational health services were helpful in supporting return to work. Support from a charity was reported to be useful for some while others found it too generic for their needs. Some participants found support from peers with similar experience of LC was helpful and provided validation.

Some people with LC accessed positive peer support via social media. However, others reported negative experiences and safety concerns regarding social media. The lack of monitoring in online forums contributed to some people feeling vulnerable to receiving incorrect information and negativity from others.

Many people with LC reported resorting to online information, with some taking this information to their GP consultation. Finally, as reported above, some turned to private healthcare in response to access issues and long waiting times, including psychological therapy, GP, medical specialties and physiotherapy.

## DISCUSSION

We explored the issue of access to community rehabilitation for LC from the perspectives of people with LC and GPs in four Scottish health boards. We identified several systemic challenges for LC service delivery which related to access, siloed services, limited resources and a perceived lack of holistic care, causing frustration for both GPs and people with LC. Similar challenges have been reported in the international qualitative literature from the perspectives of people with LC[30–32] and healthcare professionals.[33] To our knowledge, this is the first study to focus on community rehabilitation in Scotland. Although a minority of GPs expressed scepticism about LC and the need for rehabilitation and other services for this patient group, people with LC and most GPs agreed on the need for accessible, person-centred services and support, in

keeping with previous research recommending collaborative support mechanisms[17] and improved care coordination for people with LC[31] and recent findings on accessing LC rehabilitation in Canada.[34]

Regarding community rehabilitation specifically, we found that (1) some people with LC and some GPs lacked knowledge on the potential role of community rehabilitation in the management of LC; (2) having prior knowledge of rehabilitation or being a healthcare professional appeared to facilitate access to community rehabilitation and (3) people with LC who had received rehabilitation generally found it beneficial. Due to the lack of knowledge and difficulty accessing rehabilitation, however, people with LC had accessed a range of other services and sources of support, with varying success.

The negative impact of LC on the health and quality of life of our study participants is in keeping with other studies[35] and demonstrates the influence of LC on a sample living in Scotland almost 3 years on from the start of the pandemic. The persisting prevalence and impact of LC on people's lives further emphasises the need for support and services such as community rehabilitation to be available, in keeping with recommendations and research findings.[1 5 7 36 37]

People with LC should have access to personalised and multidisciplinary rehabilitation[1] and such rehabilitation is reportedly available throughout Scotland[12] delivered by a variety of service models. This study highlights some potential reasons for the mismatch between recommendations, reported service availability and low numbers of people with LC accessing community rehabilitation.[15]

Lack of GP knowledge regarding community rehabilitation and its potential role in LC may, in part, be attributed to the nature of LC as a new condition that health professionals are still learning how to manage. Previous research has reported a lack of GP understanding of the role of rehabilitation professionals in the management of conditions commonly encountered in primary care; for example, physiotherapists role in osteoarthritis management.[38] Internationally, public understanding of the role of occupational therapy has been reported as limited.[39] Therefore, in the context of a new condition with an evolving evidence base, it is perhaps not surprising that GPs and people with LC may have limited understanding of what rehabilitation professionals can offer to this patient population. The reluctance to promote the availability of services in some areas, due to pre-existing resourcing and anticipated demand-capacity issues may also have limited access to LC rehabilitation.[15]

Issues with promotion of services, clarity of pathways and interdisciplinary communication between GPs and rehabilitation professionals have been reported previously.[38 40] The findings of this study suggest that further improvements in communication and collaborative working[35] may be required to enhance access to community rehabilitation for people with LC. Indeed, the finding that prior knowledge of rehabilitation, and being a healthcare professional facilitated access to community

rehabilitation is further evidence that successfully navigating the referral system is challenging.

The findings of people with LC who had managed to access community rehabilitation being satisfied with it, and GPs wanting a LC service to refer patients to are in keeping with previous research.[19] The challenge not only lies in availability of such services, but clearly in people with LC and GPs awareness of the benefits of these services, their active promotion, and clear and timely accessibility of rehabilitation services. There is, therefore, a need to overcome the systemic challenges to accessing timely rehabilitation reported in this study. Considering the ongoing nature of living with LC, these challenges are likely to continue beyond the time and resource constraints of government funding provision for existing LC rehabilitation within rehabilitation services that were already historically underfunded and considered 'Cinderella services'.[15]

Lack of capacity in the UKs rehabilitation services is not a new phenomenon[41] but it has arguably been further highlighted since COVID-19.[42] Worldwide, there is a renewed focus on the importance of rehabilitation and the size of the unmet need.[8 43] Similar findings would arguably have emerged if this study had focused on people with other chronic conditions suitable for community rehabilitation such as fibromyalgia, ME/CFS, cardiovascular disease and stroke. While focused on LC, this study provides additional evidence of the need for increasing capacity in community rehabilitation services.[42]

### Strengths and limitations
This study explored the issue of accessing LC community rehabilitation in the Scottish context and included the perspectives of those referring and potentially being referred. One researcher conducted all interviews to ensure consistency and multiple researchers were involved in analysing and interpreting the data, including people with lived experience of LC. There are, however, some limitations.

The sample was one of convenience which limits generalisation, and although we recruited to target for the GP sample, there was under-representation from the two health boards with dedicated LC services. Therefore, the data largely represent the views of GPs from health boards where LC services were integrated into existing community rehabilitation services; it is possible that GPs views of dedicated LC services may be different.

We recruited a mostly female sample of people with LC, and mostly from health boards with integrated or a halted dedicated LC service. Their views on accessing LC services may, therefore, have been biased. Although LC has been reported as more common in females,[44] the under-representation of males in this study, as with others in the field,[17] is a limitation that needs to be addressed in future research. Our sample size was small but similar to previous studies exploring LC and its management.[17 33 36] Recruitment of people with LC was likely limited by our

reliance on social media and one LC charity; however, both mechanisms had the potential to reach many people.

We cannot claim data saturation. However, we are confident that we achieved adequate data sufficiency[41] for the findings to reflect some of the key issues within each participant group. The perspectives of men with LC and people accessing dedicated LC rehabilitation services require further exploration.[45] We conducted interviews online with one by telephone, which may have affected rapport and depth of interaction, but provided flexibility for participants and the research team. The GP data were not transcribed, which is commonly seen as a routine step in qualitative studies. Finally, participants did not check the transcripts or findings, but healthcare professionals and people with LC did comment on the findings at an open workshop.

### Implications for practice and research

There is a need for greater understanding by the public, GPs and other potential referrers of the role of community rehabilitation professionals in the management of LC. There is an equally important need for community rehabilitation services to be well promoted and accessible to the people with LC for whom they may be appropriate. LC is still a prevalent condition whose impact on individuals can be profound. The need for community rehabilitation for people with LC is likely to persist. Service providers should, therefore, consider availability and accessibility of LC rehabilitation and ensure adequate interprofessional communication and collaboration to enhance the experience for people with LC.

### CONCLUSION

We have provided further understanding of the barriers to accessing LC community rehabilitation in the Scottish context by exploring the perceptions and experiences of key stakeholders in the referral process. These findings can be used by those (re)designing community rehabilitation services for people with LC and potentially for other long-term conditions. There remains a need for greater public and GP awareness of the role of rehabilitation professionals in LC.

**Author affiliations**
[1]School of Health Sciences, Robert Gordon University, Aberdeen, UK
[2]NMAHP Research Unit, University of Stirling, Stirling, UK
[3]NMAHP Research Unit, Glasgow Caledonian University, Glasgow, UK
[4]Long COVID Scotland, Aberdeen, UK
[5]School of Health Sciences, University of Dundee, Dundee, UK
[6]Occupational Therapy, NHS Ayrshire and Arran, Irvine, UK

**Acknowledgements** We are grateful to all our participants who gave of their time to take part in interviews. We would like to thank Amanda Cardy, Northeast coordinator NRS Primary Care Network and local principal investigators Lynne Frew, Lynn Morrison and Gail Thomson-Patel for support with participant recruitment.

**Contributors** KC, ED, JHM, LA, JP, JO, AL and PS contributed to the study's conception and design. EH-W undertook data collection. ED, KC, EH-W, TT, JC, JS and ES undertook analyses and drafted the first version of the manuscript. KC led future iterations of the manuscript. All authors read and commented on the manuscript and approved the final version of it. The corresponding author attests that all authors meet authorship criteria and that nobody meeting the criteria have been omitted. ED and KC have joint responsibility for the research conduct of the study, had access to the data and controlled the decision to publish. KC has overall repsonsibility as guarantor.

**Funding** This work was supported by the Chief Scientist Office Scotland, grant number COV/LTE/20/29.

**Competing interests** All authors have completed the ICMJE uniform disclosure form at www.icmje.org/coi_disclosure.pdf and declare: all authors had financial support from the Chief Scientist Office Scotland (grant number COV/LTE/20/29) for the submitted work; ED has received research grants from NIHR and Scottish Government for the following studies: Caring for Long COVID in primary care and DBI COVID study, respectively. JHM has received funding for LOCOMOTION (Long COVID multidisciplinary consortium: optimising treatments and services across the NHS). No other relationships or activities that could appear to have influenced the submitted work are declared by any of the authors.

**Patient and public involvement** Patients and/or the public were involved in the design, or conduct, or reporting, or dissemination plans of this research. Refer to the Methods section for further details.

**Patient consent for publication** Not applicable.

**Ethics approval** This study involves human participants and ethical approval was granted from the Wales Research Ethics Committee 6 (21/WA/0118 A03), and each of the four Health Boards granted R&D management approval. Participants gave informed consent to participate in the study before taking part.

**Provenance and peer review** Not commissioned; externally peer reviewed.

**Data availability statement** Data are available on reasonable request. All data produced in the present study are available on reasonable request. Robert Gordon University holds the copyright for the full interview transcripts and may grant data sharing permission on request.

**ORCID iDs**
Kay Cooper http://orcid.org/0000-0001-9958-2511
Edward Duncan http://orcid.org/0000-0002-3400-905X
Julie Cowie http://orcid.org/0000-0002-4653-1283
Joanna Shim http://orcid.org/0000-0001-9438-9640
Emma Stage http://orcid.org/0009-0008-5309-8116
Lyndsay Alexander http://orcid.org/0000-0003-2437-9150
Jacqui H Morris http://orcid.org/0000-0002-9130-686X
Jenny Preston http://orcid.org/0000-0001-8891-2538
Paul Swinton http://orcid.org/0000-0001-9663-0696

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
