## [Reviewer comments · BMJ Open]

ARTICLE DETAILS

TITLE (PROVISIONAL)	Exploring the perceptions and experiences of community rehabilitation for Long COVID from the perspectives of Scottish General Practitioners and people living with Long COVID: a qualitative study
AUTHORS	Cooper, Kay; Duncan, Edward; Hart-Winks, Erin; Cowie, Julie; Shim, Joanna; Stage, Emma; Tooman, Tricia; Alexander, L; Love, Alison; Morris, Jacqui H.; Ormerod, Jane; Preston, Jenny; Swinton, Paul

VERSION 1 – REVIEW

REVIEWER	Yang, Yijiong Florida State University, College of Nursing
REVIEW RETURNED	26-Dec-2023

GENERAL COMMENTS	This qualitative study evaluated community rehabilitation for Long COVID based on insights from individuals affected by Long COVID and General Practitioners (GPs). The study aimed to illustrate: 1) the lived experiences of Long COVID; 2) the challenges of emergent and intricate chronic condition; 3) the systemic challenges for Long COVID service delivery; and 4) the perceptions and experiences of Long COVID and its management and rehabilitation . A few minor revisions have been suggested by the reviewer: On page 8, line 149: "A convenience sample of individuals with Long COVID (PwLC) was enlisted through the research team's social media accounts and those of their institutions, as well as through Long COVID Scotland—a volunteer-led charity managed by individuals with Long COVID." Please consider specifying the exact social media platforms used for recruiting individuals with Long COVID. On page 9, line 216: Kindly consider creating a flow chart to depict the recruitment process, indicating the number of individuals with Long COVID recruited and the reasons for exclusion. On page 10, line 230: In Table 1, regarding the years of experience in primary care, the total number of primary care physicians is not 13; one is missing. On page 45, line 23: In the attached protocol file, section 8.1 describes a different sample size estimation for statistical power compared to the information provided in the paper.
---

REVIEWER	Wade, Derick Oxford Centre for Enablement
REVIEW RETURNED	12-Jan-2024

GENERAL COMMENTS	This is a long paper describing qualitative data from an insufficient number of patients collected using a less-than-optimal method. As a much more focused short paper acknowledging the methodological weaknesses and considerable limitations and placing it in the context of decades of misunderstanding of rehabilitation and inadequate provision of services for people with chronic conditions, it might be helpful. As it stands, its length and weaknesses mean it is unlikely to have significant influence. The title is clear and accurate and probably cannot be shortened. The number of authors is large for a relatively limited study; 13 authors, interviewing 11 patients and 13 general practitioners. The journal is international, and throughout the paper, the authors should ensure that local terminology, like general practitioners, is explained so that it is understood. The use of abbreviations, particularly PWLC should be avoided The abstract is accurate but unbalanced. The conclusions do not follow from the data presented in the results section. These data do not justify the major generalised conclusions being drawn, however much one might sympathise with them and feel they are correct. The first bullet point under the strengths and limitations should be reconsidered and removed. Being first is always questionable, and being first in a relatively small and selected population does not have great significance. The remaining three bullet points are reasonable. The introduction is long – an introduction of one page of double-spaced text should be sufficient. Throughout the text, there is an impression that the authors have strong views about how things should be and wish to educate the reader. The introduction should set the context and explain why the study is needed succinctly. The method section should explain when this was undertaken. Qualitative data collection depends upon establishing good interpersonal relationships with the subjects and detecting non-verbal communication. Undertaking a qualitative study virtually must significantly reduce its validity and ability to pick up on crucial unspoken information. During the pandemic, this might be excusable, but without knowing the date, one cannot put this in a broader context. The sample was a convenience sample, not one designed to cover a broad spread (i.e. purposive). This is a significant weakness and is likely to cause significant bias. Recruitment through social media accounts adds a further level of bias. As a matter of writing and presentation, the authors use very long paragraphs, which makes it difficult to read and extract useful information. Another methodological weakness is the failure to transcribe the interviews, and the failure to refer back to the participants.
--

	Ten of the 11 participants were women, which again indicates a significant bias. Table 2 is clearly laid out, but a very large number of sub-themes are identified from a very small selected sample of patients, and it would be difficult to know to what extent the sub-themes reflect pre-existing sensitivity and ideas in the analysts sensitised to these ideas when reviewing limited amounts of data. The results section is excessively long. The first paragraph of the discussion is also long and raises many ideas that are not related to the data. The data are not put in any context. Among other things, one must recognise the chaotic and disorganised nature of rehabilitation services in general over many years and the deficiencies and vagaries of community rehabilitation, which have been repeatedly demonstrated. Precisely the same themes would arise interviewing people with multiple sclerosis living in Scotland in the community unable to access community rehabilitation. Conclusion. The authors have views about community rehabilitation, the management of long-term conditions and similar matters that they have put forward implicitly and sometimes explicitly in this paper. This study has too many flaws, including a sample that is too small, biased, and not representative or purposive to allow any conclusions to be drawn. I have not seen any comparison with other qualitative studies on this population of patients. The authors do not make a case for the scientific utility of studying a Scottish population – a case might be made, but they have not done so.
--	--

VERSION 1 – AUTHOR RESPONSE

Reviewer 1		
On page 8, line 149: "A convenience sample of individuals with Long COVID (PwLC) was enlisted through the research team's social media accounts and those of their institutions, as well as through Long COVID Scotland—a volunteer-led charity managed by individuals with Long COVID." Please consider specifying the exact social media platforms used for recruiting individuals with Long COVID.	We have added the detail that Facebook and X (formerly Twitter) were the social media platforms used.	Line 174
On page 9, line 216: Kindly consider creating a flow chart to depict the recruitment process, indicating the number of individuals with Long	We feel that a flowchart may be unnecessary due to the small numbers but can create one if required.	Lines 251-254

COVID recruited and the reasons for exclusion.	We have revised the text to enhance transparency regarding recruitment and exclusions, so it now reads: “Twenty-five PwLC expressed an initial interest in taking part in the study. Of these 25, one did not meet the inclusion criteria for this study, but they were invited to take part in the larger realist evaluation. Thirteen PwLC decided against taking part (reasons unknown).”	
On page 10, line 230: In Table 1, regarding the years of experience in primary care, the total number of primary care physicians is not 13; one is missing.	Apologies for this oversight, the data has been checked and all n(%) corrected.	Table 1 Line 269
On page 45, line 23: In the attached protocol file, section 8.1 describes a different sample size estimation for statistical power compared to the information provided in the paper.	As a qualitative study, we did not aim to recruit for statistical power, and have stated that we are unlikely to have reached data saturation (lines). Statistical power in the supplementary protocol refers to the quantitative component of the larger realist evaluation study within which this qualitative study is embedded. For this qualitative study we stated in the protocol that we would aim to recruit 3-5 community dwelling adults (in the manuscript referred to as people with Long COVID) and 2-5 GPs at each site (total 12-20 community dwelling adults and 8-20 GPs). This information can be found on Page 7 of Supp1 under the heading “Community Dwelling adults and GPs”. We therefore recruited GPs to target and under-recruited (by n=1) community dwelling adults.	NA
Reviewer 2		
This is a long paper describing qualitative data from an insufficient number of patients collected using a less-than-optimal method. As a much more focused short paper acknowledging the methodological weaknesses and considerable limitations and placing it in the context of decades of misunderstanding of rehabilitation	Thank you for these observations. We have revised the manuscript with this comment in mind, aiming for a more succinct writing style whilst authentically representing the data.	Lines 763-767

and inadequate provision of services for people with chronic conditions, it might be helpful. As it stands, its length and weaknesses mean it is unlikely to have significant influence.	We have also provided more context regarding rehabilitation services for people with chronic conditions.	
The title is clear and accurate and probably cannot be shortened. The number of authors is large for a relatively limited study; 13 authors, interviewing 11 patients and 13 general practitioners.	This study was embedded within a larger Long COVID rehabilitation realist evaluation (manuscript in process) and links to the paper published in BMJ Open in December 2023: https://doi.org/10.1136/bmjopen-2023-078740 The authors comprise study leads and co-applicants, research fellows and assistants, and people with lived experience, all of whom do fulfil the ICMJE authorship criteria by either substantially contributing to the current manuscript's conception & design OR acquisition, analysis, or interpretation of data AND all the remaining 3 criteria.	NA
The journal is international, and throughout the paper, the authors should ensure that local terminology, like general practitioners, is explained so that it is understood. The use of abbreviations, particularly PWLC should be avoided	Terminology Apologies for this oversight. We have added ("synonymous with Family Physicians/Doctors" the first time GPs are mentioned); defined clinics as outpatient/ambulatory; added 'four of the 14 Scottish territorial health boards' and replaced primary care with General Practice in Table 1. Abbreviations Our lived experience co-authors had helped in deciding to use PwLC. However, we can see that this may be interpreted as not being person-centred. In consultation with these co-authors, and consulting the literature on long COVID, we have used the abbreviation LC for Long COVID (as we feel this enhances readability of the manuscript and reduces unnecessary word count) but have removed the abbreviation PwLC (people with Long COVID) from the manuscript text. It remains as a code for the quotations, which we trust is acceptable. However, we are happy to be guided by the	Lines 110; 137; 158-159; 269 Throughout

	reviewer/editor as to use of Long COVID and people with Long COVID instead.	
The abstract is accurate but unbalanced. The conclusions do not follow from the data presented in the results section. These data do not justify the major generalised conclusions being drawn, however much one might sympathise with them and feel they are correct.	Thank you for this comment. We have revised the abstract to include more detail of the results in addition to the thematic labels and have amended the conclusions.	Lines 29-55
The first bullet point under the strengths and limitations should be reconsidered and removed. Being first is always questionable, and being first in a relatively small and selected population does not have great significance. The remaining three bullet points are reasonable.	We have revised the bullet points, which now read:  • The issue of accessing Long COVID community rehabilitation was explored from the perspectives of both potential referrers to and recipients of community rehabilitation. • One researcher conducted all interviews, ensuring consistency in their conduct. • Data were analysed and interpreted by multiple researchers, including people with Long COVID. The study is limited by the small and predominantly female sample , largely drawn from health boards with a similar approach to Long COVID rehabilitation.	Lines 59-70
The introduction is long – an introduction of one page of double-spaced text should be sufficient. Throughout the text, there is an impression that the authors have strong views about how things should be and wish to educate the reader. The introduction should set the context and explain why the study is needed succinctly.	We have revised and significantly shortened the introduction, focussing on the key points of introducing Long COVID, Rehabilitation and the focus of the study. Although we have not managed to reduce to the 1-page suggested we have endeavoured to only include essential text and hope that this is seen as an improvement.	Pages 4-6
The method section should explain when this was undertaken. Qualitative data collection depends upon establishing good interpersonal relationships with the subjects and detecting non-verbal communication. Undertaking a qualitative study virtually must significantly reduce its validity and ability to pick up on crucial unspoken information. During the pandemic, this might be excusable, but without	The manuscript states that interviews were undertaken “ during the period June 2022 to January 2023 ”. Whilst this was after lockdown restrictions had been lifted, the study had been designed to be compliant with COVID-19 restrictions. Online interviews enabled people form across a wide area of Scotland to take part and ensured flexibility in timing & scheduling (people with Long COVID could reschedule at short notice) and no need for travel by	Line 197

knowing the date, one cannot put this in a broader context.	researchers or participants (particularly important for people with Long COVID, due to fatigue and wanting to avoid personal contact, as advised by our lived experience co-authors. These reasons, and the emerging discourse on the benefits of virtual interviews for qualitative research have been summarised as: Online interviews have been utilised for some time [20]. Whilst they became a necessity during the COVID-19 pandemic [21] their potential is increasingly recognised [22]; in this study they facilitated participation across a large geographical area and flexibility for participants, whilst avoiding unnecessary social contact for people with LC.” We have also added to the limitations: “we conducted interviews online with one by telephone, which may have affected rapport and depth of interaction, but provided flexibility for participants and the research team”	Lines 199-203 Lines 798-799
The sample was a convenience sample, not one designed to cover a broad spread (i.e. purposive). This is a significant weakness and is likely to cause significant bias. Recruitment through social media accounts adds a further level of bias.	We acknowledge this limitation and have stated it more clearly in the limitations section. “The sample was one of convenience which limits generalization...”	Line 775
As a matter of writing and presentation, the authors use very long paragraphs, which makes it difficult to read and extract useful information	We have revised the entire manuscript aiming to be more succinct and using shorter paragraphs. We hope this now makes the manuscript an easier read.	Throughout
Another methodological weakness is the failure to transcribe the interviews, and the failure to refer back to the participants.	We fully acknowledge this as a limitation. We have revised the methods to read:	

	Due to the homogeneity of the GP interviews, these were not transcribed but instead were listened to on multiple occasions by two researchers, a method informed by the wider literature [23 24]. We have revised the limitations to read: “The GP data was not transcribed, which is commonly seen as a routine step in qualitative studies”	Lines 212-215 Lines 800-801
Ten of the 11 participants were women, which again indicates a significant bias.	We agree and have noted this in the limitations section and methodological bullet-points. We have added to the limitations, so it now reads: Although Long COVID has been reported as more common in females [39], the under-representation of males in this study, as with others in the field [13] is a limitation that needs to be addressed in future research.”	Lines 786-789
Table 2 is clearly laid out, but a very large number of sub-themes are identified from a very small selected sample of patients, and it would be difficult to know to what extent the sub-themes reflect pre-existing sensitivity and ideas in the analysts sensitised to these ideas when reviewing limited amounts of data. The results section is excessively long.	Table 2 reflects the Framework analysis undertaken and is included to provide an audit trail of the analysis process. However, we can see that labelling the table ‘themes’ and ‘subthemes’ is confusing and does not transparently represent the analysis undertaken. We have amended to ‘concepts,’ ‘themes’ and ‘overarching themes’ which more accurately reflects the analytical framework and should make sense to the reader when accompanied by the text in the methods (data analysis; paragraph 2).	NA

The first paragraph of the discussion is also long and raises many ideas that are not related to the data.	We have significantly revised and shortened this paragraph, removing information not directly related to the data.	
The data are not put in any context. Among other things, one must recognise the chaotic and disorganised nature of rehabilitation services in general over many years and the deficiencies and vagaries of community rehabilitation, which have been repeatedly demonstrated. Precisely the same themes would arise interviewing people with multiple sclerosis living in Scotland in the community unable to access community rehabilitation.	Thank you for this comment, with which we concur. We have provided some additional context regarding rehabilitation services.	Line 686-704
Conclusion. The authors have views about community rehabilitation, the management of long-term conditions and similar matters that they have put forward implicitly and sometimes explicitly in this paper. This study has too many flaws, including a sample that is too small, biased, and not representative or purposive to allow any conclusions to be drawn. I have not seen any comparison with other qualitative studies on this population of patients. The authors do not make a case for the scientific utility of studying a Scottish population – a case might be made, but they have not done so.	We hope that the revisions have enhanced the manuscript. We have included reference to more qualitative studies on Long COVID patients and healthcare professionals managing them within the discussion. In the introduction we describe the approach to Long COVID rehabilitation taken in Scotland and have added to the study justification: “The current study aimed to add to the growing body of knowledge by exploring people with LC and GPs’ perspectives of accessing LC community rehabilitation in the Scottish context, due to the approach to LC in this setting being distinct from other parts of the UK”	Lines 695-704 & 713-714 Lines 130-133

--	--	--

VERSION 2 – REVIEW

REVIEWER	Wade, Derick Oxford Centre for Enablement
REVIEW RETURNED	18-Mar-2024

GENERAL COMMENTS	The authors have considered the points made in my original review. They have, to a greater or lesser extent, modified the presented paper. The editor or editorial board will need to make further decisions concerning this paper because the comments I will make reflect personal preferences and the priorities of the journal in terms of the intrinsic importance of the paper and what its topic is. I will outline some of the things that concern me for the editor or editorial board to consider. Length of paper and information provided. Many qualitative studies providing much more concrete information and assistance to readers are much shorter than this paper. As a reader, I would not read this paper because it is very long for information that has low value due to the small number of subjects and the failure to separate out the problems of long Covid from general problems to do with rehabilitation and specifically community rehabilitation for any person with a long-term condition. Context. This paper is presented as if this was a unique and self-evidently important topic and fails to set this in the context. First, long Covid is probably not significantly different from any other similar disorders, such as chronic fatigue syndrome, fibromyalgia, and chronic pain. The only unique feature is the attribution by the patient or their healthcare adviser of the symptoms exclusively to a previous infection of Covid. Second, as I have commented, a survey of similar patients in Scotland with any of a large number of long-term conditions would highlight precisely the same problems, particularly if the condition were chronic fatigue syndrome or fibromyalgia. The reader needs to be informed about the similarities. If the authors believe there is something specific about long Covid, then they should draw attention to it and focus on that. Abbreviations and style. The only abbreviation I would accept is UK. There is much evidence that abbreviations reduce readability and understanding. The authors assume the reader will start at the beginning and then remember the meaning of an abbreviation. If the authors are worried about adding to the length of the paper, then they should try reducing the text by about 30%, which would be achievable and lead to a much more readable paper, more likely to be read. As an author and an editor, I know from personal experience as an author and from long experience as an editor
--

	that reductions of 30%, even after the first revision, are quite possible. It requires hard work. In terms of style, much of the method section adds material justifying what they did. In my view, the method should describe methods. Further, it is a particularly weak justification to claim that others have done the same. There are many examples of flawed measures and methods being used simply because people have done it before. It is not an adequate justification. Limitations. Although the authors now mention their limitations, they still write as if these were not limitations. Claiming that the subjects were consistent with one another is a particularly flawed justification for considering that it must be unbiased. It may well be, for example, that men have a quite different view and this is unlikely to become apparent if there is only one man. The whole reason for undertaking purpose first sampling is to check that the finding is consistent across the whole of the affected population. Consequently, the authors make claims for conclusions that do not reflect the limitations. What is the message? The paper lacks a coherent story or message. It identifies many issues but does not expand on them, and no single issue maintains the reader's interest. This is a description, not a story. There needs to be a theme running through the paper. Conclusion. The paper informs the reader what was done and what was found. A critical reader will rapidly identify the weaknesses and be able to draw their own conclusions. The reader with little interest will probably not read very far. Readers who already agree with the conclusions and findings will use this to support them. There are no hidden fatal flaws. The editor will need to decide upon its priority and the extent to which the authors should shorten the paper and adapt the style.
--	---

VERSION 2 – AUTHOR RESPONSE

Reviewer 2		
The authors have considered the points made in my original review. They have, to a greater or lesser extent, modified the presented paper. The editor or editorial board will need to make further decisions concerning this paper because the comments I will make reflect personal preferences and the priorities of the journal in terms of the intrinsic importance of the paper and what its topic is. I will outline some of the things that concern		

me for the editor or editorial board to consider. Length of paper and information provided. Many qualitative studies providing much more concrete information and assistance to readers are much shorter than this paper. As a reader, I would not read this paper because it is very long for information that has low value due to the small number of subjects and the failure to separate out the problems of long Covid from general problems to do with rehabilitation and specifically community rehabilitation for any person with a long-term condition. Context. This paper is presented as if this was a unique and self-evidently important topic and fails to set this in the context. First, long Covid is probably not significantly different from any other similar disorders, such as chronic fatigue syndrome, fibromyalgia, and chronic pain. The only unique feature is the attribution by the patient or their healthcare adviser of the symptoms exclusively to a previous infection of Covid. Second, as I have commented, a survey of similar patients in Scotland with any of a large number of long-term conditions would highlight precisely the same problems, particularly if the condition were chronic fatigue syndrome or fibromyalgia. The reader needs to be informed about the similarities. If the authors believe there is something specific about long Covid, then they should draw attention to it and focus on that.	As per Editor's comments above, we have not reduced the length of the paper for this resubmission; it has however been significantly redrafted since its first submission, including reducing the overall word count. The context is one of Long COVID, but we agree that that there are many general principles and similarities that can be drawn. We have attempted to further elaborate on this in both the introduction and discussion. Specifically, we have added: “Rehabilitation can be defined as intervention/s aimed at optimising function and reducing disability.[8] By definition, rehabilitation is multidisciplinary and applied to a range of health conditions. Whilst it is increasingly recognised that the signs and symptoms of LC share commonalities with other chronic conditions, including fibromyalgia and Myalgic Encephalomyelitis (ME)/Chronic Fatigue Syndrome (CFS)[9] and that condition-specific rehabilitation services may further fragment already scarce resources[cite], different approaches to LC rehabilitation have emerged.” “Similar findings would arguably have emerged if this study had focussed on	Lines 81-87
--	---	--------------------

Abbreviations and style. The only abbreviation I would accept is UK. There is much evidence that abbreviations reduce readability and understanding. The authors assume the reader will start at the beginning and then remember the meaning of an abbreviation. If the authors are worried about adding to the length of the paper, then they should try reducing the text by about 30%, which would be achievable and lead to a much more readable paper, more likely to be read. As an author and an editor, I know	people with other chronic conditions suitable for community rehabilitation such as fibromyalgia, ME/CFS, cardiovascular disease and stroke. Whilst focussed on LC, this study provides additional evidence of the need for increasing capacity in community rehabilitation services.[42] We also took the opportunity to cite a recently published study when discussing our findings: “To our knowledge, this is the first study to focus on community rehabilitation in the UK. Although a minority of GPs expressed scepticism about LC and the need for rehabilitation and other services for this patient group, people with LC and most GPs agreed on the need for accessible, person-centred services and support, in keeping with previous research recommending collaborative support mechanisms[cite] and improved care coordination for people with LC[31] and recent findings on accessing LC rehabilitation in Canada[34]. As per Editors comments above, we have not altered the abbreviation for Long COVID (LC).	Lines 544-547 Lines 487-492
---	--	---

from personal experience as an author and from long experience as an editor that reductions of 30%, even after the first revision, are quite possible. It requires hard work. In terms of style, much of the method section adds material justifying what they did. In my view, the method should describe methods. Further, it is a particularly weak justification to claim that others have done the same. There are many examples of flawed measures and methods being used simply because people have done it before. It is not an adequate justification. Limitations. Although the authors now mention their limitations, they still write as if these were not limitations. Claiming that the subjects were consistent with one another is a particularly flawed justification for considering that it must be unbiased. It may well be, for example, that men have a quite different view and this is unlikely to become apparent if there is only one man. The whole reason for undertaking purpose first sampling is to check that the finding is consistent across the whole of the affected population. Consequently, the authors make claims for conclusions that do not reflect the limitations. What is the message? The paper lacks a coherent story or message. It identifies many issues but	We do not consider our sample unbiased and did attempt to honestly reflect the study's limitations. We have reworked some of this section to aid clarity. Particularly we have added: "We cannot claim data saturation; however, we are confident that we achieved adequate data sufficiency [41] for the findings to reflect some of the key issues within each participant group. The perspectives of men with LC and people accessing dedicated LC rehabilitation services require further exploration." We stand by our conclusions that LC rehabilitation needs to be better understood and promoted in order for people with LC to be managed appropriately.	Lines 569-572
--	--	----------------------

does not expand on them, and no single issue maintains the reader's interest. This is a description, not a story. There needs to be a theme running through the paper.

Conclusion.

The paper informs the reader what was done and what was found.

A critical reader will rapidly identify the weaknesses and be able to draw their own conclusions. The reader with little interest will probably not read very far. Readers who already agree with the conclusions and findings will use this to support them. There are no hidden fatal flaws.

The editor will need to decide upon its priority and the extent to which the authors should shorten the paper and adapt the style.